# Integrated Transcriptome and Metabolome Analysis Reveal That Exogenous Gibberellin Application Regulates Lignin Synthesis in Ramie

Hongdong Jie [1], Long Zhao [1], Yushen Ma [1], Adnan Rasheed [1] and Yucheng Jie [1,2,*]

[1] College of Agronomy, Hunan Agricultural University, Changsha 410128, China; jhd20210218@stu.hunau.edu.cn (H.J.); azlhh@stu.hunau.edu.cn (L.Z.); mys9204@stu.hunau.edu.cn (Y.M.); adnanrasheed@hunau.edu.cn (A.R.)

[2] Hunan Provincial Engineering Technology Research Center for Grass Crop Germplasm Innovation and Utilization, Changsha 410128, China

* Correspondence: fbfcjyc@hunau.edu.cn

**Abstract:** Gibberellin regulates plant growth, development, and metabolic processes. However, the underlying mechanism of the substantial effect of gibberellin on stem height and secondary metabolites in forage ramie is unclear. Therefore, this study combined transcriptomic and metabolomics analyses to identify the mechanisms regulating growth and secondary metabolite contents in forage ramie following exogenous gibberellin application. Exogenous gibberellin application significantly reduced the lignin content in the leaves but not in the stems. At the same time, gibberellin significantly increased the total flavonoid and chlorogenic acid contents in both the stems and leaves. In addition, 293 differentially expressed genes (DEGs) and 68 differentially expressed metabolites (DEMs) were identified in the leaves. In the stems, 128 DEGs and 41 DEMs were identified. The DEGs *PER42*, *FLS*, *CYP75A*, and *PNC1* were up-regulated in the leaves, affecting phenylpropane metabolism. The joint analysis of the DEMs and DEGs revealed that the changes in the DEGs and DEMs in the leaves and stems improved the substrate efficiency in the phenol propane pathway and inhibited lignin synthesis in plants, thus shifting to flavonoid pathway synthesis. In conclusion, gibberellin treatment effectively reduces the lignin content in forage ramie while increasing the flavonoid and chlorogenic acid contents. These findings provide empirical and practical guidance for breeding for forage quality in ramie and the improvement and cultivation control of forage ramie.

**Keywords:** gibberellin; transcriptome; metabolome; lignin; flavonoids

## 1. Introduction

Ramie (*Boehmeria nivea* (L.)), which belongs to the family Urticaceae, is suited to moderate subtropical conditions. It is an important fiber crop [1] whose fresh stems and leaves contain high contents of protein, carbohydrates, ash, ether leachate, calcium, phosphorus, magnesium, and iron, making them a nutritious feed for pigs, chickens, ducks, geese, fish, and ruminants [2]. As a result, forage ramie is a good plant protein source in animal feed and an excellent source of phenolics and flavonoids [3], making it a high-quality forage for herbivorous livestock in warm climates. However, the lignin in forage ramie adversely affects the taste and digestibility of the feed. Therefore, reducing the lignin content and increasing the contents of secondary metabolites, such as flavonoids, could improve the quality of forage ramie.

Lignin, the major component in plant cell walls, is a polycyclic aromatic compound [4]. It has a complex water-insoluble chemical structure that plays an important role in maintaining plant morphological support, resistance to fungi and pathogens, and water evacuation [5]. The recalcitrance of lignin hinders the utilization of many plants in green chemistry [6]. In addition, the utilization of forage plants is limited by a high lignin content, which is the

greatest obstacle to efficient forage and crop straw feeding. The high lignin content of forage limits forage digestibility, reducing its nutritional value and palatability [7]. Therefore, one of the most effective ways of increasing forage palatability, digestibility, and nutritional value is to reduce the lignin content of plants. However, very low lignin content in plants is detrimental to plant growth and development. More importantly, a lack of lignin in the vasculature that transmits nutrients and water results in stunting and inhibition of plant growth [8]. Therefore, an appropriate reduction in lignin content without affecting normal plant growth, such as using biotechnological techniques to reduce the lignin content in forage grasses, has great potential in livestock production.

Plant hormones regulate the activities of several key enzymes in the lignin synthesis pathway, which alters the content of lignin monomers [9–12]. For example, different indole-3-acetic acid and gibberellin concentration ratios altered the lignin content in peppermint by controlling the monomeric composition of lignin in the primary bast fiber, with gibberellin stimulation also promoting the polymerization of G-type lignin [13]. Gibberellins are a class of growth regulators widely present in plants, and they regulate many physiological processes in plants, such as seed germination, stem elongation, leaf extension, and seed and fruit development [14]. Gibberellins also regulate plant growth and the synthesis of secondary metabolites, including flavonoids [15]. Notably, gibberellins increase the level of flavonoid-specific mRNAs by increasing the synthesis of anthocyanins [16]. However, the external application of gibberellins to ginkgo inhibits endogenous gibberellins, limiting the synthesis of secondary metabolites [17].

In contrast, spraying gibberellins on sweet potato stem tips significantly increases the chlorogenic acid content of stem tips after 72 h [18]. Since the precursors of lignin secondary metabolites are of the same origin as lignin monomers, the synthesis of secondary metabolites in the phenylpropanoid metabolic pathway could also be affected by the regulation of lignin monomers after hormone spraying. In ramie, the external application of gibberellin in the early stage of ramie growth promoted cell elongation and accelerated plant growth [19], which regulated the lignin synthesis rate. However, the mechanism by which gibberellin regulates flavonoid, chlorogenic acid, and lignin content in ramie remains unclear.

Herein, we compared the physiological parameters, biochemical analyses, and the expression of genes and metabolites using transcriptome data and metabolome data, respectively, in the forage ramie control (untreated) and gibberellin-treated groups. The findings in this study will help us understand how phytohormones can be used to regulate the functional components of forage ramie in production practice to improve its quality.

## 2. Results

*2.1. Effect of the External Application of Gibberellin on Growth Indicators (Plant Height and Stem Thickness) of Forage Ramie in the Field*

The plant height in the treatment group was 96.60 cm compared to 71 cm in the control group (Figure 1A). The stem thickness in the gibberellin-treated group was 10.35 mm, 21.6% higher than in the control group (8.68 mm).

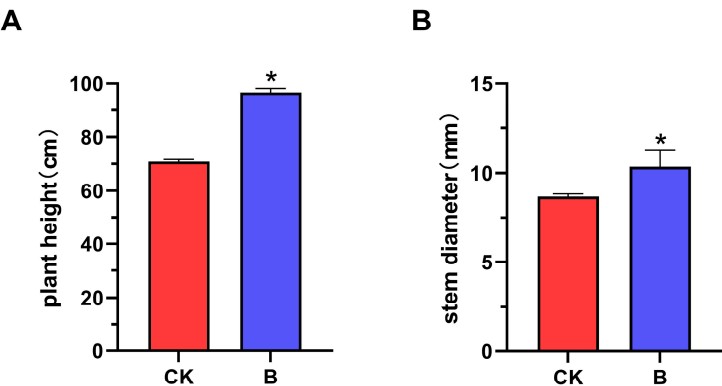

**Figure 1.** Effect of gibberellin treatment on plant height and stem thickness of forage ramie. CK indicates the control group, and B indicates gibberellin-treated group. (**A**) Plant height. (**B**) Stem thickness. The data shown are the mean and SD (error bars). * in the same column indicate significant differences at $p \leq 0.05$.

### 2.2. Effect of External Gibberellin Application on biochemical Parameters in Ramie Forage

The lignin content in the leaves was significantly decreased after gibberellin treatment (Figure 2A). Specifically, the lignin content in the leaves of the gibberellin-treated group (11.77%) was 32.5% lower than in those of the control group (15.6%). However, the total flavonoid content in the leaves of the gibberellin-treated group was significantly increased by 32.6% to 8.78 mg/g compared to in the leaves of the control group (6.62 mg/g) (Figure 2A). Similarly, the chlorogenic acid content in the leaves treated with gibberellin (8.79 mg/g) was 17.51% higher compared to in the leaves of the control group (7.48 mg/g) (Figure 2A).

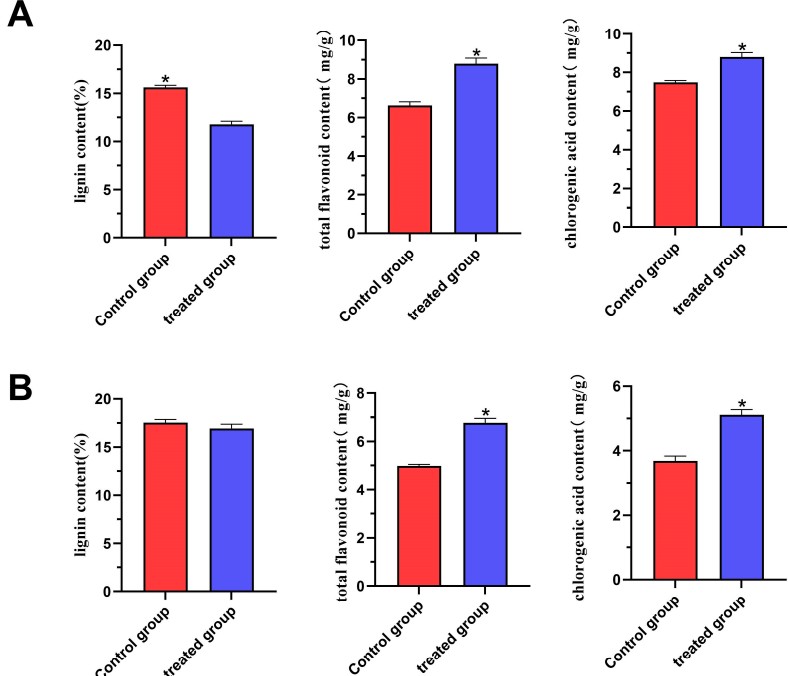

**Figure 2.** Effects of gibberellin treatment on lignin, total flavonoids, and chlorogenic acid in the (**A**) leaves and (**B**) stems of forage ramie. The data represent the mean and SD (error bars). * in the same column indicate significant differences at $p \leq 0.05$.

The stem lignin content analysis revealed that it slightly decreased by 3.4% in the gibberellin-treated ramie (17.51%) compared to the control (16.92%). However, the difference was not significant (Figure 2B). The total flavonoid content in the stems of the

gibberellin-treated group (4.98 mg/g) was significantly higher than in the stems of the control group (6.76 mg/g) (Figure 2B). In contrast, the chlorogenic acid content in the stems of the gibberellin-treated group (5.11 mg/g) was 38% higher than in the stems of the control group (3.68 mg/g) (Figure 2B). Therefore, gibberellin treatment significantly reduced the lignin content in forage ramie leaves but significantly increased the total flavonoid and chlorogenic acid contents in the leaves and stems. However, there were no significant changes in the total lignin content in the stems following gibberellin treatment.

### 2.3. Transcriptomic Analysis

#### 2.3.1. Transcriptome Sequencing and Assembly

The total base number (bp) in the three biological samples of the gibberellin-treated leaf and stem groups was 77.5–85.4 and 62.9–69.4 billion, while in the control group, it was 56.7–70.5 and 57.8–74.9 billion, respectively. After filtration of reads for quality control, the total numbers of bases in the high-quality transcriptome data from gibberellin-treated leaf and stem groups were 76.9–84.7 and 62.5–68.9 billion, respectively. The total numbers of filtered high-quality bases in the leaf and stem samples in the control group were 56.3–70 and 57.4–74.4 billion, respectively. The average Q20 was more than 97%, the average Q30 was more than 92%, and the average GC content was more than 48%, suggesting the results are valid for further analysis (Table S11).

#### 2.3.2. GO and KEGG Term Classification of DEGs

A total of 293 genes were differentially expressed in ramie leaves between the gibberellin-treated and control groups, of which 168 genes were up-regulated, and 125 genes were down-regulated. In addition, 128 differentially expressed genes (DEGs) were identified in the ramie stem samples, including 30 up-regulated genes and 98 down-regulated genes in the gibberellin-treated samples (Table S1). These DEGs in ramie stem and leaves may be important genes related to lignin and flavonoid biosynthesis.

#### 2.3.3. Enrichment Analysis of the DEG Functions in GO Annotation and KEGG Pathways

Gene Ontology (GO) annotation analysis of the DEGs in the leaves (CK-YvsB-Y) revealed many DEGs were related to cellular processes, metabolic processes, cells, organelles, binding sites, and catalytic activity (Figure 3A, Table S2). Among the DEGs between the control and gibberellin-treated stems (CK-JvsB-J), the cell process, metabolic process, cell, cell site, organelle, and binding site were the most common GO terms (Figure 3B, Table S2).

A

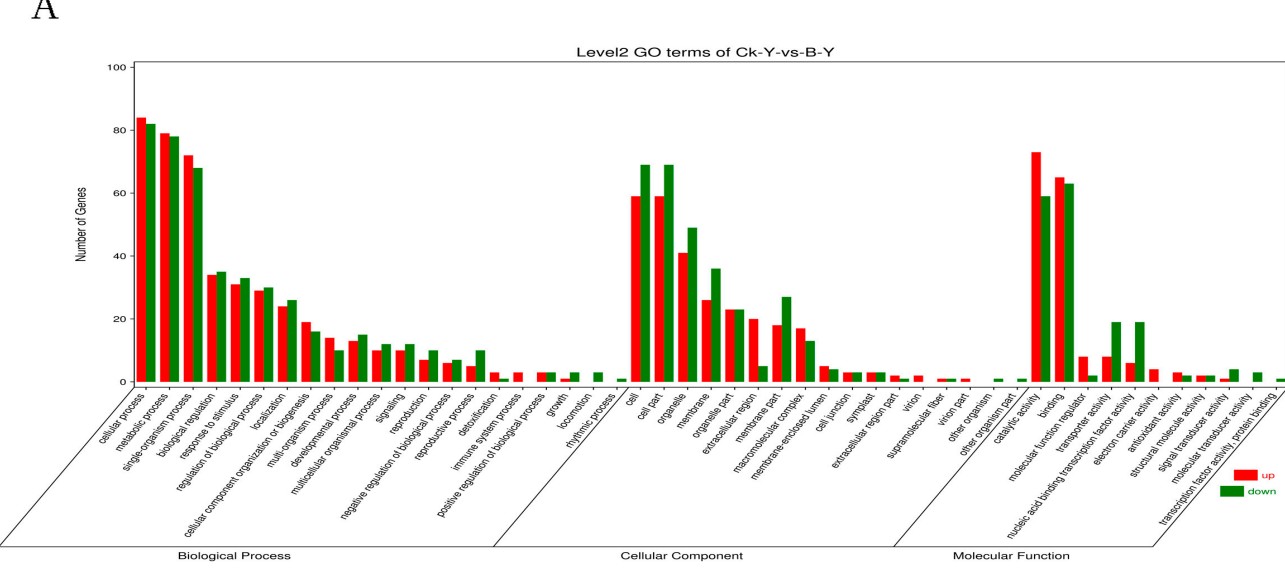

**Figure 3.** *Cont.*

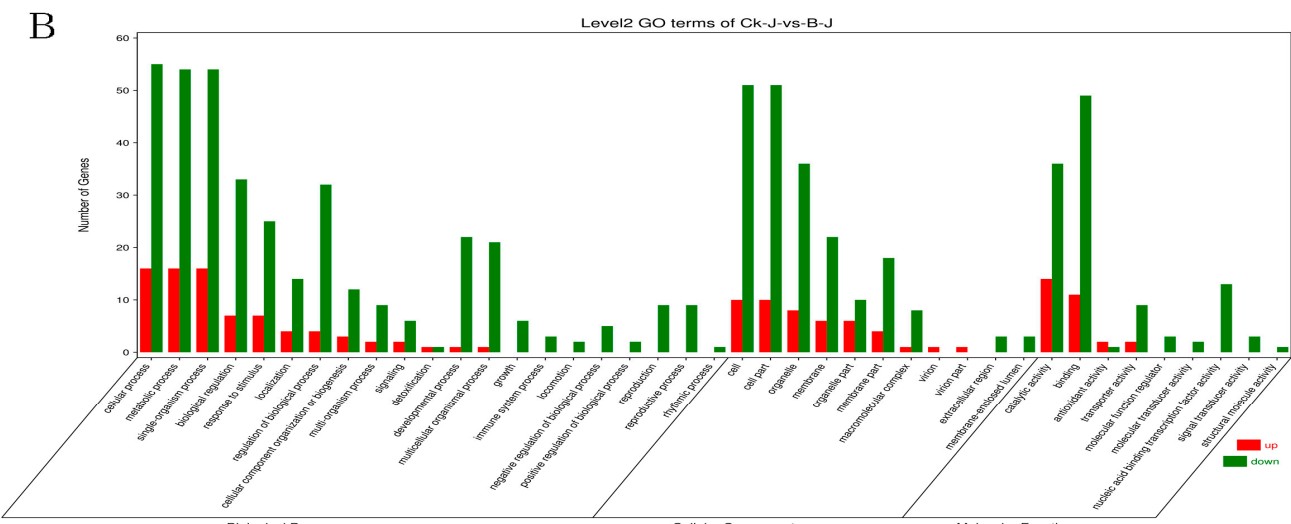

**Figure 3.** Gene Ontology (GO) enrichment and classification histogram. (**A**) The GO functional enrichment analyses of the differentially expressed genes (DEGs) in the leaves between the control and gibberellin-treatment groups (CK-YvsB-Y). (**B**) The GO functional enrichment analyses of the DEGs in the stem between the control and gibberellin-treatment groups (CK-JvsB-J). The ordinate is the number of the DEGs annotated with the term, while the abscissa is the secondary GO term. Up- and down-regulated genes in the gibberellin-treated group are indicated in red and green, respectively.

The GO functional enrichment analysis of the DEGs revealed that the first five GO terms significantly enriched in forage ramie leaves were the extracellular region (GO:0005576), glucosyltransferase activity (GO:0046527), gibberellin 2-β-dioxygenase activity (GO:0045543), xylan transferase activity (GO:0016762), and urea binding (GO:0033219) (Figure 4A). Meanwhile, the first five GO terms significantly enriched in the forage ramie stem were stem shoot system development (GO:0048367), plant organ development (GO:0099402), leaf development (GO:0048366), thallus development (GO:0048827), and secondary bud formation (GO:0010223) (Figure 4B). Overall, the most significant GO terms in the leaf and stem tissues were completely different.

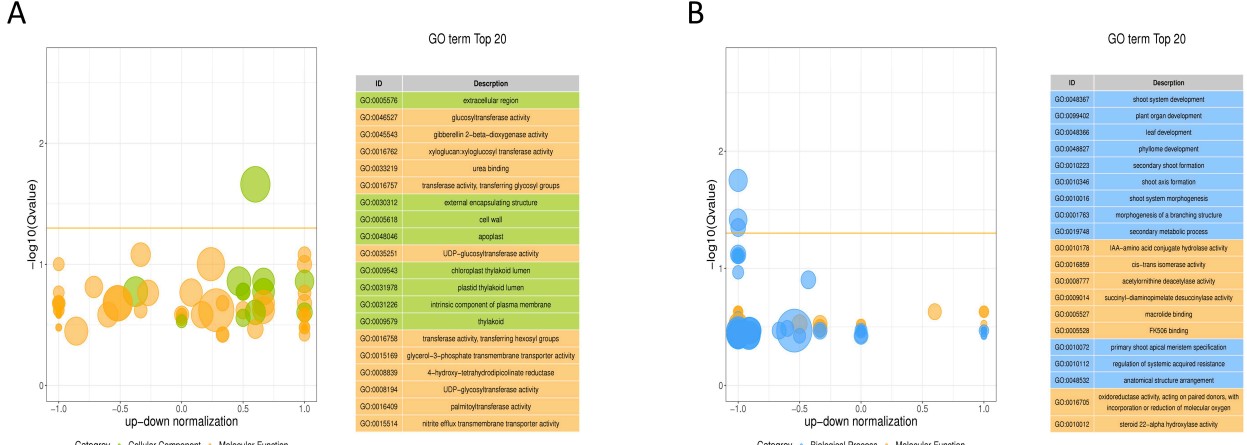

**Figure 4.** The Gene Ontology (GO) functional enrichment analysis bubble diagram of the differentially expressed genes (DEGs) in the leaves (CK-YvsB-Y) and stem (CK-JvsB-J). (**A**) The top 20 GO terms in the leaves (CK-YvsB-Y). (**B**) The top 20 GO terms in the stem (CK-JvsB-J). The ordinate is the $-\log_{10}(p\text{-value})$ in the GO functional enrichment bubble chart, and the larger the ordinate, the smaller the *p*-value. The abscissa is the ratio of the difference between the number of up-regulated and down-regulated genes to the total DEGs. The orange line indicates the significance threshold level at *p* = 0.05, and the bubbles above the threshold line are significant.

The Kyoto Encyclopedia of Genes and Genomes enrichment analysis revealed that the biosynthetic pathways of secondary metabolites were significantly enriched in both the leaves and stems in the gibberellin-treated group compared to the control group. However, the pathways significantly enriched in the leaves and stems were different. In the leaves, nitrogen metabolism, photosynthetic antenna proteins, diterpene biosynthesis, biosynthesis of secondary metabolites, metabolic pathways, glycosphingolipid biosynthesis of isoglobulin series, flavonoid biosynthesis, glycosphingolipid biosynthesis of lactic acid and new lactic acid series biosynthesis of cutin, gelatin and wax, and glutathione metabolism were the most significantly enriched pathways (Figure 5A, Table S3). In contrast, zeatin biosynthesis, isoflavone biosynthesis, glutathione metabolism, glucosinolate biosynthesis, secondary metabolite biosynthesis, brassinosteroid biosynthesis, metabolic pathway, diterpene biosynthesis, and unsaturated fatty acid biosynthesis were the most significantly enriched pathways in stems (Figure 5B, Table S3). The phenylpropane and flavonoid biosynthesis were significantly enriched in both the leaves and stems. Therefore, the DEGs in these two metabolic pathways could have an important effect on lignin synthesis in plants. The KEGG pathway enrichment in the leaves and stems is presented in the Supplementary Materials (Tables S4 and S5).

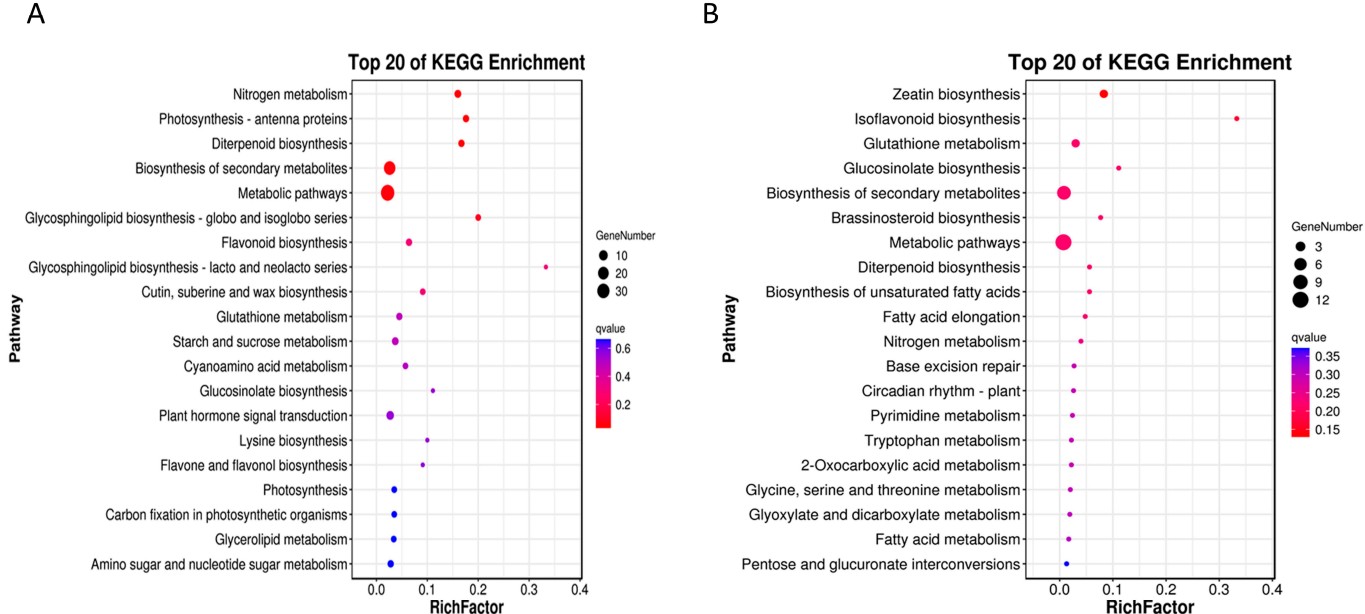

**Figure 5.** Kyoto Encyclopedia of Genes and Genomes (KEGG) pathway enrichment bubble map. (**A**) The KEGG pathway enrichment map of the differentially expressed genes (DEGs) in the leaves between the control and the gibberellin-treated groups. (**B**) The KEGG pathway enrichment map of the DEGs in the stem between the control and the gibberellin-treated groups. The pathway names are shown along the ordinate, and the abscissa indicates the enrichment factor. The larger the abscissa, the more reliable the enrichment and significance of the gene products in the pathway. Bubble size represents the number of genes; larger bubbles represent more genes.

### 2.3.4. qRT-PCR Validation of DEGSs

Comparing the expression levels of the nine DEGs obtained using qRT-PCR and the fragments per kilobase of transcript per million mapped reads (FPKMs) values of the transcriptome sequencing data revealed that the expression estimates of each gene in transcriptome sequencing were consistent between the two methods, suggesting that the transcriptome data are reliable (Figure 6).

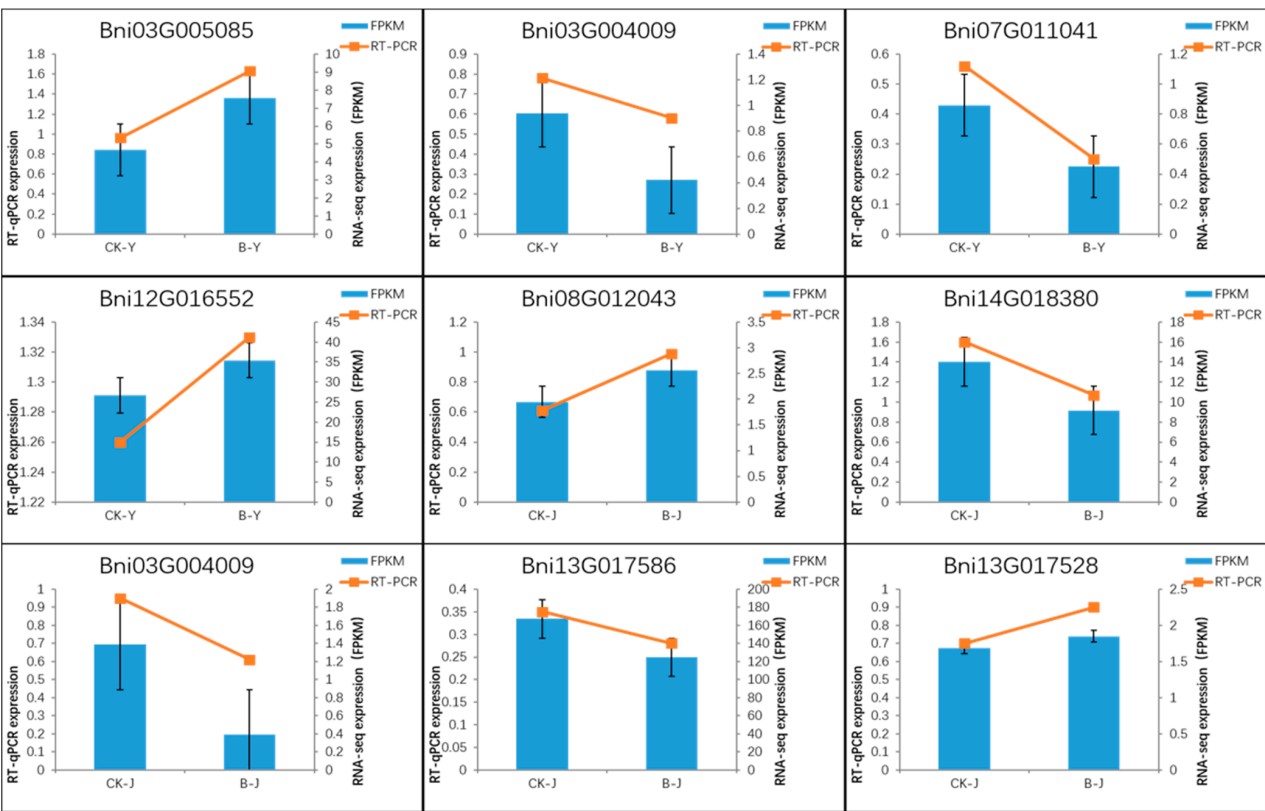

**Figure 6.** Nine differentially expressed genes in the phenylpropane pathway based on the fragments per kilobase of transcript per million mapped reads (FPKMs) and qRT-PCR detection results.

*2.4. Metabolomics*

2.4.1. Effects of Gibberellin on the Metabolites in Leaves and Stems of Forage Ramie

The principal component analysis (PCA) score scatterplot of the control and gibberellin-treated ramie revealed the metabolic differences between the leaves and stems of the two treatment groups (Figure 7). PC1 and PC2 explained 44.8% and 39.9% of the variance, respectively, among samples.

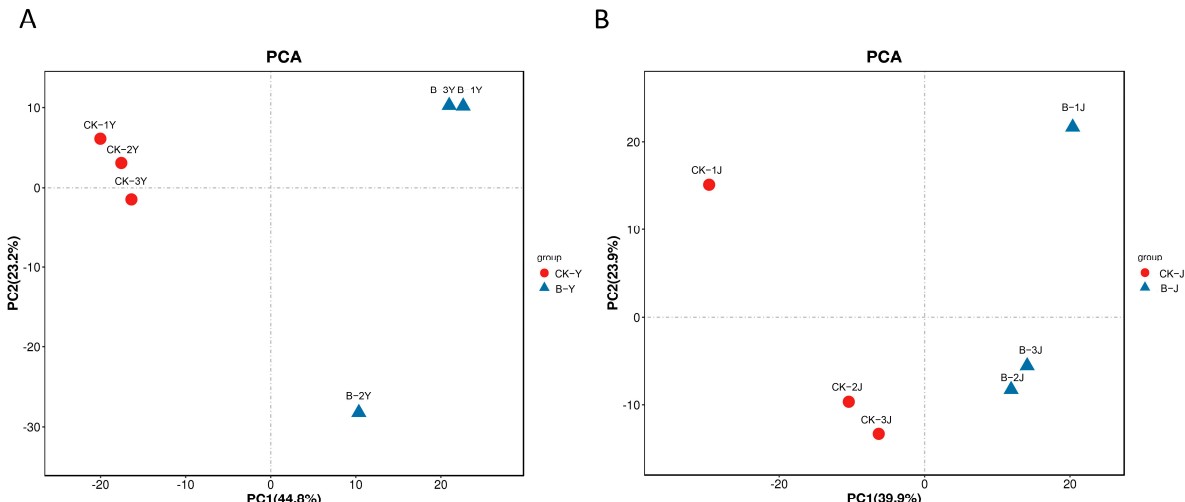

**Figure 7.** Principle component analysis (PCA) score scatterplot. (**A**) PCA of the control and gibberellin-treated leaves. (**B**) PCA of the control and gibberellin-treated stems. Red represents the control sample (CK-Y, CK-J), and blue indicates the gibberellin-treated group (B-Y, B-J). The shorter the distance between samples within the same group, the better the repeatability.

A total of 68 differentially expressed metabolites in the leaves and 41 in the stems were identified between the control and gibberellin-treated ramie. Among them, 23 differential metabolites in the leaves were up-regulated, and 45 were down-regulated (Figure 8A, Table S6). In the stem, 11 differential metabolites were up-regulated, while 30 were down-regulated (Figure 8B, Table S7).

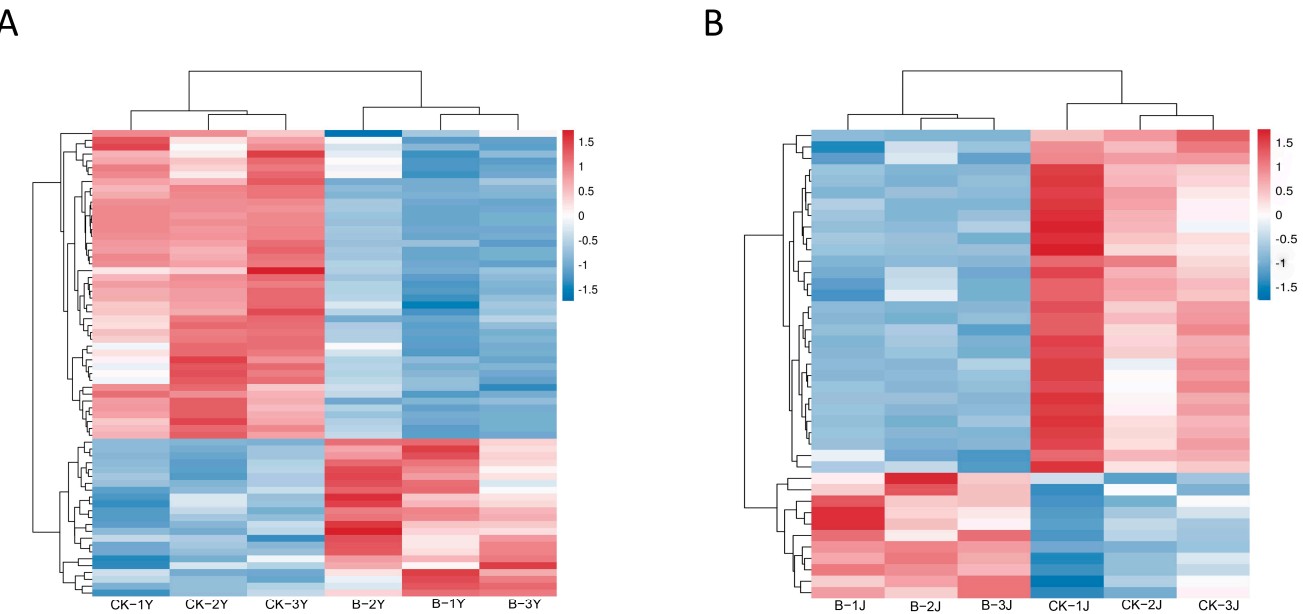

**Figure 8.** Heat map and metabolite clustering using the z-scores.

### 2.4.2. KEGG Pathway Enrichment Analysis of the Differential Metabolites

The KEGG pathway enrichment analysis revealed the effect of gibberellin and ethephon on metabolites in terms of forage ramie growth (Figure 9). In the leaves, the differential metabolites were mainly concentrated in the flavonol and flavonol biosynthesis, carbapenem biosynthesis, arginine and proline metabolism, aminoacyl-tRNA biosynthesis, and unsaturated fatty acid biosynthesis, and the alanine, aspartic acid, and glutamate metabolism pathways, among other pathways (Figure 9A, Table S8). The differential metabolites enriched in these metabolic pathways possibly affect the polymerization of lignin during plant growth and the synthesis of secondary metabolites in plants. In the stem, the differential metabolites were mainly concentrated in the unsaturated fatty acid biosynthesis, citric acid cycle (TCA cycle), flavonoid biosynthesis, 2-oxycarboxylic acid metabolism, glycine, serine and threonine metabolism, and β-alanine metabolism pathways, among other pathways (Figure 9B, Table S9). Overall, the flavonoid and flavonol biosynthesis pathways were enriched in gibberellin-treated ramie leaves and stems, thus significantly increasing the production of secondary metabolites in forage ramie. This promotes the formation of key enzymes catalyzing the valorization of lignin to flavonoids, isoflavones, and anthocyanins, thus reducing the lignin content and increasing the concentration of plant secondary metabolites such as flavonoids.

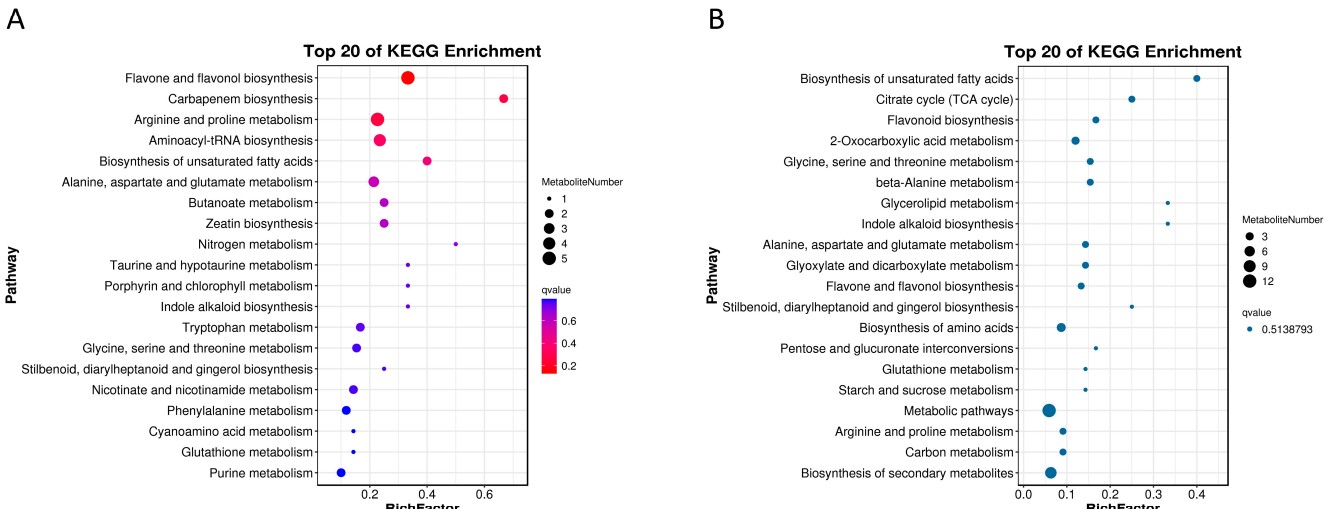

**Figure 9.** Kyoto Encyclopedia of Genes and Genomes pathway enrichment bubble diagram of the differential metabolites between the gibberellin-treated ramie and the control. (**A**) KEGG pathway enrichment analysis of the differential metabolites in leaves. (**B**) KEGG pathway enrichment analysis of the differential metabolites in the stem.

### 2.5. Combined Metabolomic and Transcriptomic Analysis

Transcriptome analysis revealed that nitrogen metabolism, biosynthesis of secondary metabolites, photosynthesis of antenna proteins, diterpene biosynthesis, metabolic pathways, glycosphingolipid biosynthesis of isoglobulin series, flavonoid biosynthesis, glycosphingolipid biosynthesis of lactic acid and new lactic acid series, degradation of cutin, gelatin and wax biosynthesis, and glutathione metabolism were the significantly enriched pathways in gibberellin-treated leaves. In the stems, zeatin biosynthesis, isoflavone biosynthesis, glutathione metabolism, glucosinolates biosynthesis, secondary metabolite biosynthesis, brassinosteroid biosynthesis, metabolic pathway, diterpene biosynthesis, unsaturated fatty acid biosynthesis, and fatty acid extension were the most significantly enriched pathways. On the other hand, metabolomic data revealed 68 differential metabolites enriched in 42 metabolic pathways in the leaves, while 41 differential metabolites enriched in 38 metabolic pathways were detected in stems. The flavonoid synthesis and biosynthesis of secondary metabolites were most abundant in stems and leaves. The joint metabolome and transcriptome analysis of differential genes and metabolites in the phenylpropane biosynthesis pathway revealed the general increase in phenylpropane and flavonoid biosynthesis (Table S10). Precisely, spraying gibberellin on forage ramie up-regulated the differential genes affecting phenylpropane metabolism in leaves, such as *PER42*, *FLS*, *CYP75A*, and *PNC1*, but down-regulated neochlorogenic acid (5-*O*-caffeoylquinicacid), a key differential metabolite in the leaves. Overall, the synthesis of secondary metabolites in the phenylpropane metabolic pathway was inevitably affected. At the same time, the abundance of flavonoid metabolites increased. However, the up- and down-regulation of the two key genes, *PER42* and *PNC1*, which are the precursors of lignin synthesis, ultimately affected lignin synthesis (Figure 10). The DEGs affecting phenylpropane metabolism in the stems included the down-regulation of *4CLL1* and *PNC2*, with the down-regulation of 5-*O*-caffeoylquinicacid and up-regulation of 1-*O*-sinapoyl-D-glucose metabolites. Collectively, the down-regulation of these differential genes and metabolites decreased the lignin content in the stems (Figure 11).

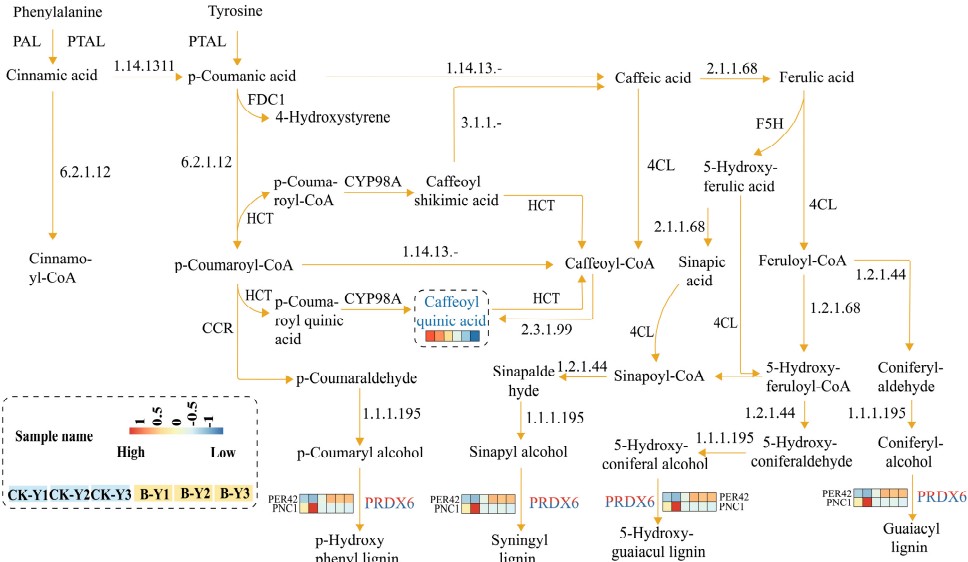

**Figure 10.** The biosynthetic pathway of phenylpropane in ramie leaves. The differential genes are indicated as a heat map next to the relevant enzymes; the dashed boxes indicate the differential metabolites.

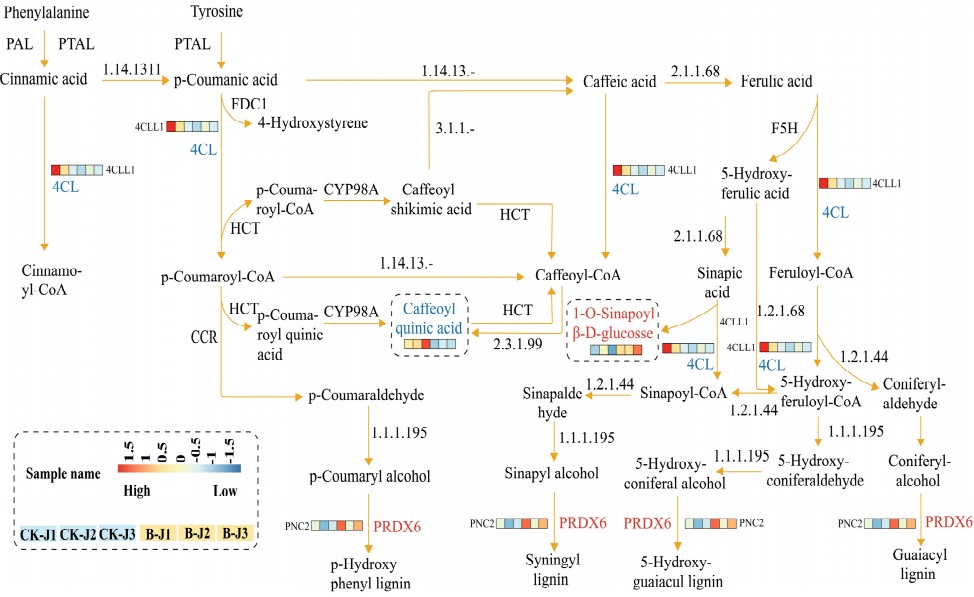

**Figure 11.** The biosynthetic pathway of phenylpropane in a ramie stem. The differential genes are indicated as a heat map next to the relevant enzymes; the dashed boxes indicate the differential metabolites.

## 3. Discussion

Forage ramie is harvested early for whole-plant feeding. Still, even its relatively low lignin content at this time affects its palatability and digestibility. Therefore, reducing the lignin content or changing its composition without affecting normal plant growth and development is crucial for the development of the ramie industry. Exogenous spraying of gibberellin promotes plant growth and development while increasing the total contents of phenols and flavonoids in the roots, stems, and leaves of gibberellin-treated plants [20]. Gibberellin regulates flavonoid biosynthesis [21], with a lower gibberellin content promoting flavonoid synthesis in the leaf [22]. At the same time, lignin and secondary metabolites in plants are regulated by specific metabolic compounds and gene expression. Plant hormones are naturally occurring small organic signal molecules, indispensable in plant stress resis-

tance and maintaining crop yield and nutritional quality [23]. This study comprehensively analyzed the metabolome and transcriptome of forage ramie to establish how forage ramie regulates lignin and flavonoids in response to gibberellin treatment. This is the first report on the metabolomic and transcriptomic regulation of the external application of gibberellin on forage ramie. The findings in this study provide a rationale for further research on the mechanism and genetic improvement in lignin regulation in response to gibberellin treatment in forage ramie.

The phenylpropane pathway is the starting point for the synthesis of flavonoids, coumarins, and lignins [24]. The first three steps in the phenylpropane pathway are catalyzed by three enzymes, phenylalanine ammonia-lyase (PAL), cinnamic acid 4-hydroxylase (C4H), and 4-coumaryl-CoA ligase (4CL), the precursors for all downstream secondary metabolites [25]. PAL is a key and rate-limiting enzyme [26] that acts as the gate in the phenylpropane metabolic pathway. Lignin biosynthesis depends on PAL expression in plants. On the other hand, C4H is a cytochrome P450-dependent monooxygenase that catalyzes the hydroxylation of cinnamic acid to p-coumaric acid, the second step in the phenylpropane pathway [27]. The third key enzyme, 4CL, connects the phenylpropionic acid pathway with the lignin-specific biosynthesis pathway. It mainly catalyzes the production of cinnamate coenzyme, an ester, from cinnamic acid, and it is also the metabolic flow regulating enzymes in the synthesis of lignin and other phenylpropane compounds [28]. It catalyzes the production of the corresponding CoA ester, an essential downstream element in the plant phenylpropane pathway. The final production of flavonoids, tannins, flavonoids, lignin, and other metabolites [29] is achieved via downstream processes in the common phenylpropane pathway, where the lignin monomer biosynthesis pathway occurs. It parallels other branch phenylpropane pathways, such as the flavonoid pathway [30]. Therefore, the regulation of lignin content is likely to directly affect the biosynthesis of functional components, such as flavonoids.

In this study, ramie plant height and stem thickness were significantly increased after gibberellin treatment. The plant height in the gibberellin-treated group was 96.60 cm compared to 71 cm in the control group, a 36.05% difference. The stem thickness in the gibberellin-treated group (10.35 mm) was 21.6% greater than that in the control group (8.68 mm). Thus, gibberellin treatment promoted ramie growth and development. Moreover, the lignin content in the leaves of the gibberellin-treated group (11.77%) was 32.5% lower than in those of the control group (15.6%). In contrast, the total flavonoid content in the leaves of the gibberellin-treated group (8.78 mg/g) was significantly increased by 32.6% compared to the control group (6.62 mg/g). Similarly, the chlorogenic acid content in the leaves of the treatment group (8.79 mg/g) was 17.51% higher than in those of the control group (7.48 mg/g). In the stem, the lignin content in the treatment group (17.51%) was decreased by 3.4% compared to the control (16.92%), although the difference was not significant. In addition, the total flavonoid content in the stems of the treatment group (4.98 mg/g) was 26.3% higher than in those of the control group (6.76 mg/g). In contrast, the chlorogenic acid content in the stems of the gibberellin-treated group (5.11 mg/g) was 38% higher than that in the stems of the control group (3.68 mg/g). This implies that exogenous gibberellin inhibited lignin synthesis in ramie but increased the accumulation of other secondary metabolites. This phenomenon was also previously found in ramie, where the decrease in lignin content increased the anthocyanin content, making its leaves more palatable [31].

The transcriptomic and metabolomic analysis of gibberellin-treated forage ramie revealed that different DEGs and differential metabolites regulated the biosynthesis of metabolites in the leaves and stems. Specifically, the DEGs *PER42*, *FLS*, and *CYP75A*, which regulate phenylpropane metabolism, were up-regulated in leaves, while *PNC1* was down-regulated. The up-regulation of *FLS* and *CYP75A* down-regulated neochlorogenic acid (5-*O*-caffeoylquinic acid), a differential metabolite in the leaves. Flavonol is synthesized through the flavonoid pathway, catalyzed by flavonol synthase (*FLS*) in plants. The overexpression of *CmFNS* and *CmFLS* promotes the biosynthesis of flavonoids and flavonols, which

inhibits anthocyanin accumulation in plants while increasing the abundance of flavonoid metabolites. At the same time, the up-regulation of two key genes, *PER42* and *PNC1*, which function upstream of lignin synthesis, ultimately affected lignin synthesis.

In the stems, the DEGs affecting phenylpropane metabolism included the down-regulated *4CLL1* and *PNC2*, resulting in the down-regulation of 5-*O*-caffeoylquinicacid and the up-regulation of 1-*O*-sinapoyl-D-glucose metabolites. The down- and up-regulation of these differential genes and metabolites down-regulated the lignin content in the stems, suggesting that the down-regulation of 5-*O*-caffeoylquinicacid affects lignin synthesis.

Chlorogenic acid synthesis is a complex process. *HCT* is important for chlorogenic acid synthesis in many plants [32]. For example, silencing the *NbHCT* gene in tobacco increases the chlorogenic acid content in stem tissue. In addition, the inhibition of *AtHCT* gene expression in *Arabidopsis thaliana* increases flavonoid accumulation while decreasing lignin accumulation [33]. Based on the leaf and stem transcriptome and metabolome data, HCT in the phenylpropane pathway may be the key enzyme promoting plant flavonoid and chlorogenic acid biosynthesis. Thus, the down-regulation of *HCT* may reduce lignin synthesis, resulting in the accumulation of flavonoid and chlorogenic acid content in plants. Additionally, the knockout of *HCT* reduces lignin content and elevates the flavonoid concentration in *Arabidopsis* [34]. At the same time, silencing *HCT* inhibits lignin biosynthesis while the abundance of flavonoids is increased, which redirects the metabolic flux to flavonoid biosynthesis, highlighting the key function of *HCT* in lignin biosynthesis [35].

Furthermore, the transcriptome analysis identified the key functional genes in lignin formation, systematically outlining how exogenous gibberellin spraying regulates lignin accumulation in forage ramie. Overall, this forms a theoretical basis for gibberellin application on forage ramie.

## 4. Materials and Methods

### 4.1. Plant Material, Experimental Site, and Handling Methods

The experimental site was located at Hunan Agricultural University (Changsha, China). The 'Zhong Ramie No. 1'. variety of *Boehmeria nivea* was used as the experimental material. Fifteen seedlings exhibiting uniform growth were randomly selected per treatment and labeled, with three replicates per treatment. Treatment plots were separated in the experimental site to prevent mutual interference during spraying between treatments. Each treatment was divided into a cell, randomly arranged, and repeated three times, with six ramie plants per cell. Application of fertilizer (urea 23.5 kg/667 m$^2$ and potassium chloride 15 kg/667 m$^2$) was carried out on the same day the forage ramie was cut, 27 August 2021. No fertilizer was applied during the forage, growth, or development stages of ramie. On the tenth day after mowing, the ramie height was measured using a straight ruler from the base of the stem to the top of the stem. In addition, the stem diameter was also measured halfway from the base using Vernier calipers.

Gibberellin (China Beijing Solarbio Technology Co., Ltd., Beijing, China) was diluted to 10 mg/L and sprayed on ramie 10 days after it was cut. The control was sprayed with distilled water. The whole plants were sprayed using a hand-pressed small spray can. Ramie was treated with gibberellin twice at an interval of seven days. Sampling was carried out when 80% of the forage ramie reached 80 cm. Before sampling, the side rows were removed from each plot. Fifteen plants per plot with consistent growth were randomly selected based on the five-point sampling method. The plants were cut, and the stems and leaves were separated with a sharp knife. Each experimental sample was divided into two groups. One group was dried at 105 °C for 10 min and then baked at 65 °C to a constant weight to determine the lignin, total flavonoid, and chlorogenic acid contents. The other group was frozen in liquid nitrogen and preserved at −80 °C for transcriptome sequencing and metabolite analysis.

### 4.2. Determination of Lignin, Chlorogenic Acid, and Total Flavonoid Contents

#### 4.2.1. Determination of Lignin Content

The dried samples were crushed and screened through a 0.2 mm sieve. Next, the lignin content per sample was determined using a Solarbio lignin determination kit (BC4200, Solarbio, Beijing, China) following the manufacturer's instructions. Each sample was replicated thrice.

#### 4.2.2. Determination of Total Flavonoids

The dried samples were crushed and screened through a 0.2 mm sieve. Subsequently, the total flavonoid content in the samples was determined using the Solarbio Total Flavonoids Kit (BC1335) following the manufacturer's instructions. Each sample was replicated thrice.

#### 4.2.3. Determination of Chlorogenic Acid Content

A ≥98% chlorogenic acid standard sample was sourced from Hefei Bomei Biotechnology Co., Ltd. (Hefei, China). Briefly, 20 mg of the chlorogenic acid standard sample was dissolved in 50% ethanol (0.040 mg/L) in a reagent bottle and increased to a total volume of 100 mL. Next, 20 mL of the resulting solution was concentrated to 10 mL with 50% ethanol. The standard curve was then generated by absorption of the chlorogenic acid standard solution at 0 to 1 ppm. Briefly, 9 mL of the chlorogenic acid standard solution was placed in a 10 mL bottle, and 50% ethanol was added before shaking. Finally, the regression equation for determining the chlorogenic acid content in ramie samples was established at a 310 nm wavelength.

The dried ramie samples were ground using a plant grinder and passed through a 60-mesh screen. Next, 0.1 g of the ground sample was placed in a 20 mL calibrated test tube and soaked in ethanol with pH 4 (solid/liquid, 1:50) for 24 h before the ultrasonic-assisted extraction for 30 min at 40 °C, filtration, and washing. Extraction was repeated twice, and the filtrate was combined to produce 10 mL of filtrate for chlorogenic acid content estimation. Subsequently, 1 mL of the filtrate was diluted six-fold (not exceeding the maximum measure of the standard curve), and the absorbance value at 310 nm was measured using 50% ethanol as the reference. Finally, the absorbance value (A) was converted into a concentration (C) (mg/mL) using the standard curve equation as follows:

$$\text{Chlorogenic acid yield} = (C \times V_1 \times V)/(m \times 1000) \times 100\% \tag{1}$$

$C$—concentration of chlorogenic acid after dilution (mg/mL), $V_1$—constant volume of extract (mL).

$V$—dilution multiple, $m$—raw material output of ramie sample.

### 4.3. RNA Extraction, Library Construction, and Sequencing

The total RNA from the ramie frozen samples was extracted using the RNAprep Pure Plant Kit (DP441, Tiangen, Beijing, China) according to the manufacturer's instructions. Next, the quality of the extracted total RNA was assessed using a NanoPhotometer spectrophotometer (IMPLEN, Westlake Village, CA, USA), and checked using RNase-free agarose gel electrophoresis, a Qubit 2.0 fluorometer (Life Technologies, Carlsbad, CA, USA), and Agilent Bioanalyzer 2100 system (Agilent Technologies, Santa Clara, CA, USA). After total RNA was extracted, eukaryotic mRNA was enriched using Oligo(dT) beads.

The first strand of cDNA was synthesized using the M-MuLV reverse transcriptase system. Briefly, the RNA chain was degraded by RNase H, and the second chain cDNA was synthesized using DNA polymerase. Next, the double-stranded cDNA was ligated to the sequencing adapter before amplification. Approximately 200 bp cDNA fragments were purified via screening using AMPure XP beads. The cDNA was amplified. Finally, the cDNA library was obtained and sequenced on the Illumina Novaseq6000 system at Chideo Technology Co., Ltd. (Guangzhou, China).

The reads obtained after sequencing were filtered to remove adapters and low-quality bases and obtain high-quality clean reads using fastp (version 0.18.0) according to the following criteria: removal of reads with adaptors; removal of reads containing more than 10% unknown nucleotides; removal of low-quality reads containing more than 50% low-quality bases (Q value $\leq$ 20).

The genes with $p < 0.05$ and $|\log_2 (\text{folding change})| > 1$ were defined as the differentially expressed genes (DEGs) between the control and gibberellin-treated groups.

### 4.4. Metabolite Analysis

4.4.1. Sample Preparation and Extraction

Biological samples (ramie) were freeze-dried using a vacuum freeze dryer (Scientz-100F; Scientz, Ningbo, China). The freeze-dried samples were crushed using zirconia beads in a mixed mill (MM 400 mm, Retsch, Haan, Germany) at 30 Hz for 1.5 min. Next, 100 mg of the freeze-dried powder was dissolved in 1.2 mL of 70% methanol solution, swirled six times every 30 min and for 30 s each time, and then incubated at 4 °C overnight in the refrigerator. After centrifuging for 10 min at 12,000 rpm, the extracts were filtered (SCAA-104, 0.22 μm X ANPEL, Shanghai, China, (http://www.anpel.com.cn/; accessed on 22 September 2022) before UPLC-MS/MS analysis.

4.4.2. Data Preprocessing and Metabolite Identification

The compounds extracted were analyzed using an LC-ESI-MS/MS system (UPLC, Shimpack UFLC SHIMADZU CBM30A, http://www.shimadzu.com.cn/ accessed on 22 September 2022); MS/MS (Applied Biosystems 6500 QTRAP, http://www.appliedbiosystems.com.cn/; accessed on 22 September 2022). First, 2 μL samples were injected onto a Waters ACQUITY UPLC HSS T3 C18 column (2.1 mm × 100 mm, 1.8 μm) operating at 40 °C and with a flow rate of 0.4 mL/min. The mobile phases used were acidified water (0.04% acetic acid) (Phase A) and acidified acetonitrile (0.04% acetic acid) (Phase B). Compounds were separated using the following gradient: 95:5 Phase A/Phase B at 0 min; 5:95 Phase A/Phase B at 11.0 min; 5:95 Phase A/Phase B at 12.0 min; 95:5 Phase A/Phase B at 12.1 min; 95:5 Phase A/Phase B at 15.0 min. The effluent was connected to an ESI–triple quadrupole–linear ion trap (Q TRAP)–MS.

Following UPLC-MS/MS analysis, Analyst1.6.1 software was used for data filtering, peak detection, alignment, and calculation. Next, the metabolites were identified by querying their UPLC-MS/MS spectra data against the internal and public databases (MassBank, KNApSAcK, HMDB [36], MoToDB, and METLIN [37]) and comparing m/z values, RT, and fragmentation patterns with standards. The variable importance in the projection (VIP) score in the (O) PLS model was used to rank the most distinguishable metabolites between the two treatments. The log(mean VIP)threshold was set to 1. In addition, the differential metabolites were screened using univariate analysis (*t*-test). Thresholds of $p \leq 0.05$ and VIP $\geq$ 1 were considered as significant metabolic differences between the two treatments.

We performed qualitative and quantitative analyses of metabolites in biological samples and screened for differential metabolites between groups.

### 4.5. Validation of Real-Time quantitativePCR (qRT-PCR)

Nine differentially expressed genes were selected to verify the reliability of the transcriptome data. Each reaction system contained 20 μL, consisting of 10 μL of SYBRgreen-MasterMix, 0.4 μL of forward and reverse primers, 2 μL of cDNA, and 7.2 μL of ddH$_2$O. The specific primers were designed using Primer5.0 software. The amplification conditions were as follows: initial denaturation at 95 °C for 15 s, 60 °C for 30 s, 72 °C for 30 s, and 40 cycles.

The relative expression of each gene was calculated using the $2^{-\Delta\Delta CT}$ method, with *actin* as the internal reference gene [38]. The primer sequences are shown in Table S10.

*4.6. Statistical Analysis*

All quantitative data were analyzed using SPSS 22.0 (IBM Corp., Armonk, NY, USA). Bioinformatics analysis of transcriptome sequencing and metabolic data was carried out using the R statistical computing environment.

**5. Conclusions**

The analysis of field characters, transcriptome, and metabolomics of forage ramie treated with gibberellin versus the control revealed the mechanism of reducing lignin content while increasing the flavonoid and other secondary metabolites in forage ramie in response to hormone regulation. When lignin biosynthesis is inhibited, the metabolic flux promotes flavonoid biosynthesis. These findings provide an empirical basis for forage ramie breeding and comprehensive utilization of Ramie varieties.

**Supplementary Materials:** The following supporting information can be downloaded at: https://www.mdpi.com/article/10.3390/agronomy13061450/s1. Table S1: Differentially expressed gene classification. Table S2: GO enrichment analysis. Table S3: KEGG enrichment analysis of differentially expressed genes. Table S4: CK-Y VS B-YPhenylpropanoid biosynthesis and Flavonoid biosynthesis. Table S5: CK-J VS B-JPhenylpropanoid biosynthesis and Flavonoid biosynthesis. Table S6: Ck-Y-vs-B-Y.Differentially expressed metabolites. Table S7: Ck-J-vs-B-J.Differentially expressed metabolites. Table S8: Ck-Y-vs-B-Y. differential metabolite enrichment pathway table. Table S9: Ck-J-vs-B-J.differential metabolite enrichment pathway table. Table S10: Primer list of differentially expressed genes. Table S11: Throughput and quality summary of RNA-sequences.

**Author Contributions:** H.J. conducted the experiment, analyzed the data, and wrote the manuscript. A.R., Y.M. and L.Z. reviewed and improved the manuscript. Y.J. supervised the study. All authors have read and agreed to the published version of the manuscript.

**Funding:** This study was financially supported by the National Natural Science Foundation of China (32071940), the China National Key R&D Program (2019YFD1002205-3 and 2017FY100604-02), and the Foundation for the Construction of Innovative Hunan (2020NK2028). Special Project for Grass Planting and Straw Processing and Utilization in Hunan Province's Herbivorous Animal Industry Technology System (2019–2023). National Crop Germplasm Resource Bank Feed Fiber Dual Use Crops and Grass Germplasm Resource Branch Project (2019–2023).

**Institutional Review Board Statement:** Not applicable.

**Informed Consent Statement:** Not applicable.

**Data Availability Statement:** Not applicable.

**Acknowledgments:** Thanks to lab members who assisted with the research work.

**Conflicts of Interest:** The authors declare no conflict of interest.

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
