# Peer review of "Integrated Transcriptome and Metabolome Analysis Reveal That Exogenous Gibberellin Application Regulates Lignin Synthesis in Ramie"

_agronomy, doi:10.3390/agronomy13061450_

Round 1

Reviewer 1 Report

Overall comments:

The topic is relevant since it addresses the scientific bases for the use of phytohormones, in this case, gibberellin, in the regulation of functional components in forage ramie production and for improvement of the quality, supported with appropriate references.

The conclusions based on the comparison of physiological parameters, biochemical analyses, and the expression of genes and metabolites through the use of transcroptome and metabolome, are consistent with the evidence presented and suited to the main objective indicated.

Results

Use the same type of letter for the word Figure as in the text of the article

Figure 1, indicate what CK and B mean. It could be Control group and Treated group

2.2 Effect…..

The lignin…………treatment (Fig. 2A)…… (Fig. 2C)… (Fig. 2E).

The stem lignin…….(Fig. 2B)…….(Fig. 2D)…….(Fig. 2F)

Figure 2, there is no need for using so many letters, just use A for Leaves and B for stems, and change CK-J for Control group and B.J for the treated group

2.3. Transcriptomic Analysis 116

2.3.1. Transcriptome sequencing and assembly

The total……further analysis (Table S11)

2.3.3. Enrichment analysis of the DEGs function in the KEGG Pathway

GO…….activity (Fig. 3A, Table S2). Among.....(Fig. 3B, Table S2).

The GO functional…….. (Fig. 4A). At the same….

Figure 4, no need to use parenthesis for the letters A and b. Change b for B.

The KEGG…….(Fig. 5B, Table S3). The phenylpropane…..(Tables S4 and S5).

Figure 5, use the same type of letter. There is no need to use parenthesis for A and B.

2.3.4. q RT-PCR analysis of the differential gene expression

Comparing…..(Fig. 6).

Figure 6. Nine differential expressions of the …….

Figure 7, use the same type of letter. There is no need to use parenthesis for A and B.

Figure 8, use the same type of letter. There is no need to use parenthesis for A and B. this Figure is not mentioned in the text of the article.

2.4.2. KEGG pathway enrichment analysis of the differential metabolites

The KEGG pathway enrichment….(Fig. 9). In the leaves, the…..(Fig. 9A, Table S8). The differential….(Fig. 9B, Table S9). Overall, the…..

Figure 9, use the same type of letter. There is no need to use parenthesis for A and B.

Figure 9. KEGG pathway…and the control, (A) KEGG…….in leaves, (B) KEGG……….

2.5. Combined metabolome and transcriptome analysis

Transcriptome analysis…..lignin synthesis (Fig. 9). The differential…..content in the stems (Fig. 10).

Figure 10, use the same type of letter.

Figure 11, use the same type of letter.

Materials and methods

Indicate what kg/mu mean in the first citation

4.2.3 determination of chlorogenic acid content

It is not necessary to use two decimals in 20.00, 9.00, 0.00, and 1.00

Check the spelling

Use h instead of the word hours

4.3 RNA extraction , library construction, and sequencing

The methodology is not explained in detail

4.5. Validation of qRT-PCR

Check the redaction and spelling

Authors contribution

Check the redaction and spelling

References

There is no consistency in the way the references are written. Just check the first four.

References 1, 31, 32, 37, 38, and 39 are not mentioned in the text of the article.

Reference 40 is not in order since it is mentioned after reference 18

References  9 and 10 are mentioned out of order.

The first letter of words that make up the name of a journal must be in uppercase letter, for example in references 4, 7, 8, 12, 16, 21, 24, 25, 26, 30, 36, 37, 38, and 40.

The space between words, and words and punctuations is not properly used.

References lack the DOI, for example: Reference 40: BMC Genomics 21, 40 (2020). https://doi.org/10.1186/s12864-020-6457-8

redaction must be improved as well as the punctuations.

Author Response

Thank you for your comments. We have revised it according to your opinion.

Reviewer 2 Report

This study combined transcriptomic and metabolomics analyses to establish the mechanisms regulating growth and secondary metabolite contents in forage ramie following exogenous gibberellin application, which was mainly involved with flavonoid metabolism and phenylpropane metabolism. After my review, I found there are some problems that need to be corrected or clarified in the manuscript.

1) The manuscript lacks information on the LC-MS instrument analysis for metabolomics. It would be helpful to include details such as the device model, column, and detection conditions used.

2) The statistical method used to analyze the difference in gene/metabolite abundance is unclear. While ANOVA appears to have been used based on Figures 1 and 2, for two-group comparisons, it is more common to use the student's t-test. Additionally, significant differences (p<0.05) are typically denoted with an asterisk, rather than the alphabets "a" and "b" used by the authors.

3) The KEGG pathway enrichment analysis of differently accumulated metabolites only yielded a few statistically significant pathways (Table S8 and S9), and the expected pathways of flavonoid metabolism and phenylpropane metabolism were not among them. This may be due to the limited number of metabolites detected and annotated. Therefore, it would be more appropriate to only include the significant pathways in the figures and further discussions.

It is always helpful to have native speakers review and polish the English in a manuscript to ensure clarity and accuracy.

Author Response

(The authors gave the same response as above.)
